# The Rationale and Current Status of Endotoxin Adsorption in the Treatment of Septic Shock

**DOI:** 10.3390/jcm11030619

**Published:** 2022-01-26

**Authors:** Jakub Śmiechowicz

**Affiliations:** Department of Anaesthesiology and Intensive Therapy, Wroclaw Medical University, Borowska 213, 50-556 Wroclaw, Poland; jakub.smiechowicz@umw.edu.pl

**Keywords:** endotoxin, septic shock, blood purification, bacterial translocation

## Abstract

Lipopolysaccharide, the main component of the outer membrane of Gram-negative bacteria is a highly potent endotoxin responsible for organ dysfunction in sepsis. It is present in the blood stream not only in Gram-negative infections, but also in Gram-positive and fungal infections, presumably due to sepsis-related disruption of the intestinal barrier. Various pathways, both extra- and intracellular, are involved in sensing endotoxin and non-canonical activation of caspase-mediated pyroptosis is considered to have a major role in sepsis pathophysiology. Endotoxin induces specific pathological alterations in several organs, which contributes to poor outcomes. The adverse consequences of endotoxin in the circulation support the use of anti-endotoxin therapies, yet more than 30 years of experience with endotoxin adsorption therapies have not provided clear evidence in favor of this treatment modality. The results of small studies support timely endotoxin removal guided by measuring the levels of endotoxin; unfortunately, this has not been proven in large, randomized studies. The presence of endotoxemia can be demonstrated in the majority of patients with COVID-19, yet only case reports and case series describing the effects of endotoxin removal in these patients have been published to date. The place of blood purification therapies in the treatment of septic shock has not yet been determined.

## 1. Introduction

Sepsis is recognized as a global health problem with an estimated nearly 50 million cases and 11 million deaths recorded worldwide in 2017, representing almost 20% of all global deaths [1]. Recent meta-analysis of epidemiological evidence, related to the burden of hospital-acquired sepsis, showed mortality between 30.1% and 64.6% among ICU-treated patients [2]. Septic shock is characterized by persistent hypotension requiring vasopressor support and a serum lactate level > 2 mmol/L, despite adequate fluid resuscitation. In the continuum of sepsis severity it carries the worst prognosis, with mortality reaching up to 92% in some studies [3].

In addition to standard therapy, which includes infection control (antibiotics, controlling the source), cardiovascular resuscitation (administering fluids, vasoactive agents), and organ support, modulation of the host response is assumed to improve outcome, with low-dose corticosteroids being most commonly advocated [4,5]. An alternative approach includes extracorporeal therapies aimed at removing molecules that are involved in the immune reaction to invading microorganisms. Endotoxin plays a prominent role in the pathogenesis of sepsis, and the idea to neutralize its detrimental capacities continues to attract the attention of researchers and clinicians. The aim of this review is to summarize the current knowledge on the pathophysiology of endotoxin and the existing evidence on the efficacy of extracorporeal blood purification treatment relative to the adverse impact of endotoxin on organ function.

## 2. Endotoxin, Lipopolysaccharide

In 1892, Richard Pfeiffer introduced the concept of endotoxin to define the phenomenon that a toxic substance—an insoluble part of the bacterial cell—evokes a typical picture of bacterial infection, even without the presence of living bacteria [6]. Many years of research were needed to determine the exact structure, function, and mechanism of the action of lipopolysaccharide that proved to be responsible for this effect.

Lipopolysaccharide (LPS) is the major component of the cell wall of Gram-negative bacteria, comprising roughly 75% of the surface of the outer leaflet of the outer membrane of the cell wall. LPS is a glycolipid; it consists of a hydrophobic lipid part, called lipid A, which is anchored in the outer leaflet, and a hydrophilic polysaccharide part, which extends outside the cell. The polysaccharide part is composed of two domains: the core oligosaccharide and the O antigen. The O antigen (also called the O-chain) is a polysaccharide which is composed of several oligosaccharide units and is bound to lipid A through the core region [7]. LPS molecules create a tight hydrophobic structure with strong bonds that form a permeability barrier that protects the bacterial cell against antimicrobial factors [8]. With a few exceptions, for example, *Treponema pallidum*, LPS is produced by most Gram-negative bacteria [9].

Although the general LPS structure is conserved, many differences are possible among species of bacteria. An LPS molecule without the O-chain which is produced by some species of Gram-negative bacteria is referred to as “rough” LPS, as opposed to a “smooth” LPS, which includes the O antigen [10,11]. LPS is essential for bacterial survival in a hostile environment and Gram-negative bacteria that lack the LPS or that have the LPS without an O-chain are more sensitive to the host’s defense mechanisms and antibiotics [8].

Lipid A deserves particular attention, as this part of the LPS molecule is sensed by the host and is responsible for activating the immune system and the toxic and pyrogenic effects of endotoxin. The structure of lipid A synthesized by various Gram-negative bacteria can differ in the number and the length of fatty acid chains attached and the presence or absence of phosphate groups or other residues [8]. The variable structure of lipid A determines the stimulatory or inhibitory action. Lipid A with a hexa-acyl structure, i.e., when the diglucosamine backbone has two phosphates and six fatty acyl chains attached, is best sensed by the host’s complex of myeloid differentiation factor 2 and the toll-like receptor 4 (MD2-TLR4) [12].

Lipopolysaccharide in the cell membrane of anaerobic Bacteroidales, a predominant phylum in the commensal microbiota of the human gut, has an under-acylated (tetra- or penta-acyl) lipid A in its structure and is a potent TLR4 inhibitor. Consequently, it silences TLR4 pathway signaling, thus facilitating the host’s tolerance of gut microbes [13]. It is unknown whether this phenomenon has any effect on the progression of sepsis [14]. In fact, the lipid A structure of many Gram-negative bacteria, including Pseudomonas aeruginosa, does not have six FA chains [12]. Yersinia pestis has the ability to produce hexa-acyl LPS at 21–27 °C and tetra-acyl LPS at 37 °C, and thus it escapes the host’s first line of defense in fleas and mammals. A genetically modified strain of Y. pestis which produces hexa-acylated LPS at 37 °C appeared avirulent, as it facilitated the early recognition of infection and the effective onset of immune signaling [15]. During chronic infection, alterations in the LPS molecule are possible and are thought to facilitate the evasion of host immune defenses and biofilm adaptation [16].

Gram-negative bacteria constitute a major part of the gut microbiota and are a source of LPS that is estimated to possibly exceed 1 g [17,18]. In health, minor amounts of LPS can translocate into the bloodstream with the potential to trigger an immune response. In order to protect the host from noxious over-activation of the immune system, several mechanisms exist for LPS detoxification [19,20]. LPS that enters the bloodstream is rapidly sequestered by lipoproteins, mainly high density lipoproteins (HDL) in cooperation with the phospholipid transfer protein (PLTP). Lipoproteins transport LPS to the liver, where it is inactivated by enzymes acyloxyacyl hydrolase and alkaline phosphatase and excreted in the bile [19]. 

Another mechanism relies on the binding of LPS to HDL3, the small form of HDL. LPS from the intestines is captured locally by HDL3 particles that are produced by intestinal epithelial cells to form the HDL3-LBP-LPS complex. This complex hides LPS from being detected by liver macrophages, and instead leads to the inactivation of LPS by the plasmatic enzyme acyloxyacyl hydrolase (AOAH), thus protecting the liver from inflammation and fibrosis that may develop in the course of chronic LPS exposure [21]. 

The capacity of the detoxifying mechanisms is insufficient when sepsis-related disruption of the intestinal barrier occurs, and an increased amount of endotoxin enters the blood stream. This is likely when the intestinal epithelium, formed by only one layer of cells with a huge area of approximately 30 m^2^, is compromised by hypoperfusion, inflammation, dysregulation of commensal flora, or sepsis-induced ileus resulting in increased gut-barrier permeability [22,23,24,25].

## 3. Lipopolysaccharide Sensing Pathways

Lipopolysaccharide is sensed via extracellular and intracellular pathways that lead to the activation of the immune response.

### 3.1. Toll-like Receptor 4–Myeloid Differentiation Protein 2 (TLR4-MD-2) Pathway

The toll-like receptor 4 (TLR4) is the main sensing receptor for LPS, and it is one of the pattern recognition receptors responsible for the early detection of invading microbes by the innate immune system. TLR4 is expressed on the surface of macrophages, monocytes, neutrophils, dendritic, and epithelial cells, as well as within endosomes, forming the front line of the host’s defense against Gram-negative bacteria. LPS molecules in the bacterial cell wall and also soluble LPS-aggregates are dissociated and bound by LPS Binding Protein (LBP), carried to form a complex with either a soluble or membrane bound cluster of differentiation-14 (CD14), and subsequently transferred to the toll-like receptor 4/myeloid differentiation-2 (MD-2) complex, which promotes the TLR4/MD-2 dimerization necessary for activating intracellular MyD88-dependent and TRIF-dependent pathways. Both pathways lead to the production and release of pro-inflammatory cytokines and type I interferones (IFNs), respectively [26,27,28]. Immune hyperactivation from the inappropriate triggering by pathogens and the cytokine storm leads to organ damage, multi-organ failure, and death [29]. 

The progress in research on LPS recognition systems, witnessed in the last decade, led to important discoveries of TLR4-independent LPS-sensing pathways that may have a central role in the pathophysiology of sepsis and related mortality.

### 3.2. Transient Receptor Potential (TRP) Ion Channels

Transient receptor potential ion channels are membrane-bound channels that serve as cellular sensors of environmental and intracellular stimuli. LPS sensing by TRP channels has been demonstrated in neurons and airway epithelial cells [30,31]. The activation of TRPA1 channels in nociceptive neurons by the LPS of pathogenic bacteria generates pain during inflammation [32]. Activation of the TRPV4 channels in the airway epithelium boosts ciliary beat frequency and the production of bactericidal nitric oxide, which facilitates the pathogen clearance from the airways. LPS sensing by TRP channels provides an immediate response to invading pathogens, which is faster and independent of the canonical TLR4 immune pathway [31].

### 3.3. Intracellular LPS Sensing

The activation of caspases plays a crucial role in intracellular pathogen detection and defense. LPS can enter the cytosol as LPS/outer-membrane-vesicle (OMV)-high mobility-group-box-1 (HMGB1) complexes internalized through a receptor for advanced glycation (RAGE). LPS that enters the cytoplasm of macrophages, as well as endothelial and epithelial cells, is sensed by inflammatory caspases—caspase-11 in mice and caspase-4/5 in humans—and leads to the induction of pyroptosis, an inflammatory form of cell death. Activated caspases cleave gasdermin D, which causes pore formation in the cell membrane with subsequent cell lysis and the release of proinflammatory IL-1β and IL-18 [33]. Inflammasome activation and pyroptosis are important mechanisms of the innate immune defense against pathogens that are capable of invading the cytosol and play a major role in sepsis pathophysiology. Caspase-11 has been found to be responsible for bacterial clearance in Klebsiella pneumoniae and Acinetobacter baumannii, as well as Burkholderia lung infections [33]. It is speculated that caspases may be responsible for sensing penta-acylated LPS, which is not detected by TLR4 [34]. Caspase-mediated pyroptosis of endothelial cells has a fundamental role in the host’s defense and immune surveillance functions of the microvasculature [35]. Excessive activation of pyroptosis causes extensive cell death and immense inflammation leading to organ failure and septic shock [36].

## 4. Organ Damage Caused by Sensing Endotoxin

Endotoxin plays a very prominent role in the pathogenesis of sepsis. It is one of the most important pathogen-associated molecular patterns (PAMP), and a large burden of endotoxin triggers an excessive, uncontrolled systemic inflammatory response that leads to multi-organ failure and death. Moreover, endotoxin induces specific pathological alterations in several organs that contribute to the outcome (Figure 1).

### 4.1. The Kidney

Acute kidney injury (AKI) develops in at least 40–50% of patients with sepsis or septic shock and is associated with significantly higher mortality [37,38,39,40]. In addition to septic alterations, AKI presents with metabolic and fluid abnormalities, necessitating adjustments in volume therapy and pharmacotherapy, most notably limiting antimicrobial choice. The pathophysiology of septic AKI is complex and, in addition to hypoperfusion, interactions between vascular, tubular, and inflammatory factors are involved. Although the exact mechanism underlying renal dysfunction in sepsis remains unknown, there is strong experimental evidence supporting the prominent role of the toll-like receptor 4 (TLR-4), which is expressed in the kidney [41]. Its activation causes cytokine and chemokine release; leukocyte infiltration, which results in endothelial dysfunction; tubular dysfunction and altered renal metabolism and circulation [42]. TLR-4 receptors are located in the tubular epithelium and in the glomeruli and vascular endothelium. Endotoxin is filtered in renal glomeruli, internalized by S1 proximal tubules through TLR4 receptors, and interactions between endotoxin and S1 tubules result in severe oxidative stress and damage to the neighboring S2 segments [42,43]. TLR4 directly inhibits bicarbonate absorption in the medullary-thick ascending limb, downregulates renal sodium, chloride, and glucose transporters, causes luminal obstruction, and reduces tubular flow, among other effects [42]. Endothelial activation and alterations to glomerular glycocalyx and the deposit of NETs in kidney tissue secondary to endotoxic shock also contribute to kidney injury [44,45]. Direct renal damage by endotoxin can explain the occurrence of AKI in sepsis, even when hemodynamic parameters are well-controlled [43]. In fact, protocolized hemodynamic resuscitation did not influence either the development or the course of AKI in patients with septic shock [40]. As a result, the concept of equating sepsis-induced AKI to acute tubular necrosis, attributed to ischemia from hemodynamic changes, has been replaced by the theory of the interplay between inflammation and oxidative stress, microvascular dysfunction, and the adaptive response of the tubular epithelial cells to the septic insult [46].

### 4.2. The Lung

In mice subjected to LPS-induced sepsis, pronounced histological alterations in the lungs were found, with thickening of the septum, edema, congestion, and high leukocyte infiltration into the interstitium, which correlated with a significant increase in the serum concentrations of NETs and the extent of lung injury [45]. In another experimental study, lung injury was attributed to LPS-triggered pyroptosis of the endothelial cells in the lungs; LPS sensing in the endothelial cytoplasm via caspase-4/5/11-mediated pyroptosis led to disruption of the endothelial barrier resulting in pulmonary edema, the release of pro-inflammatory cytokines, fluid protein leakage, and a massive influx of leukocytes [35]. The pyroptotic response was augmented when the expression of caspase-4/5/11 was enhanced by concomitant priming with extracellular LPS via LPS binding to TLR4 [35].

### 4.3. The Heart

Toll-like receptors 4 are expressed in cardiomyocytes and their activation elicits an inflammatory response with the production of cytokines and chemokines with a negative effect on cardiac contractility [47]. In healthy volunteers, endotoxemia resulted in a reduction in the left ventricular ejection fraction and an increase in the left ventricular end diastolic volume [48]. In mice, LPS administration resulted in significant pathological changes in the myocardial bundles, congestion of the capillaries with the presence of leukocytes attached to the endothelium, and pathological changes in the cardiomyocytes seen upon histological examination [45]. The results of other studies indicated that sepsis-associated cardiac dysfunction was also mediated by mechanisms other than TLR4 [49].

### 4.4. The Liver

The liver is an important participant in the body’s reaction to endotoxemia. Murine studies demonstrated that endotoxin uses both TLR4 and caspase-11/gasdermin D (GsdmD) pathways to induce the release of HMGB1 from hepatocytes—the major source of circulating HMGB1 in sepsis [50]. Complexes of hepatocyte-released HMGB1 and LPS are delivered via RAGE into the cytosol of macrophages and endothelial cells, where LPS activates caspase-11 and induces pyroptosis and cell death [51]. The intracellular LPS-sensing pathway is considered to have a central role in the pathogenesis of sepsis [33]. 

In the liver, LPS affects the architecture of the sinusoidal endothelium and blood flow velocities, which leads to extravasation of neutrophils and neutrophil–hepatocyte interactions, decreases protein S and thrombomodulin synthesis, which contributes to a pro-coagulant state and has a direct cytotoxic effect on hepatocytes [44,52]. In mice subjected to LPS-induced endotoxemia, histological changes in the liver included enlarged sinusoids, an increased volume of endothelial cells with rounded nuclei, a high number of leukocytes in the lumen, Kupffer cell hypertrophy and hyperplasia, along with the presence of leukocytes close to periportal areas and congestion of the central vein with swollen hepatocytes [45].

### 4.5. The Vascular Endothelium

Endothelial cell dysfunction is thought to be the key factor in the progression from sepsis to organ failure [44]. The presence of endotoxin in the blood causes shedding of the glycocalyx lining of the vascular endothelium that leads to the loss-of-barrier function, the formation of edema, and the dysregulation of vascular tone, among other effects [44]. LPS triggered, caspase-dependent pyroptosis in endothelial cells results in disruption of the endothelial barrier, fluid leakage, and the development of ALI [35].

## 5. Endotoxin Removal

In patients with septic shock, an early start of an appropriate therapy is generally advised. The SSC guidelines underscore the importance of the prompt implementation of bundles of care, including cultures, broad-spectrum antibiotics, fluids, vasopressors, and lactate levels [5]. Yet, despite an optimal treatment, the mortality caused by septic shock remains high. The idea that the endotoxin burden triggers a detrimental, excessive, and uncontrolled response of the immune system that leads to multi-organ damage and death is the mainstay of anti-endotoxin therapies.

Endotoxemia is a frequent finding in patients with septic shock and correlates with increased organ failure and higher mortality [53,54,55]. In a group of 157 patients with septic shock and endotoxemia on admittance to the ICU, the group with endotoxin activity above 0.4 had a much higher mortality than the group of patients with low endotoxin activity, regardless of the infecting microorganism [56]. In postsurgical patients admitted to the ICU, high levels of endotoxin were associated with a longer ICU length of stay [57]. In septic shock patients, endotoxemia occurs frequently in the absence of Gram-negative bacteria and is also found in patients with documented Gram-positive and fungal infections [53,56,58,59,60].

The presence of endotoxin in systemic circulation is presumed to be due to the increased translocation of bacteria and their toxins as a result of a failure of the gut barrier. Changes in the tight junction, indicating intestinal barrier dysfunction and increased permeability, can occur as early as 1 h following sepsis [61]. Significant endotoxemia is also present in patients after prolonged cardiopulmonary bypass [62] and other conditions that result in splanchnic hypoperfusion [63,64].

Several therapies targeting endotoxin in sepsis were studied. Anti-endotoxin antibodies [65,66], phospholipid emulsion [67], bactericidal/permeability-increasing protein (BPI) [68], and synthetic lipid A antagonist [69], among others, failed to demonstrate any benefit in septic patients. 

The first attempts at extracorporeal elimination of endotoxin from solution were reported in the 1970s. Research that led to the development of the PMX cartridge, which contains an immobilized polymyxin B—a polypeptide antibiotic with the capacity to neutralize endotoxin, began in 1981 in Japan. Clinical trials with PMX were started in 1989 and the treatment became widely used in Japan after 1994, upon receiving approval from the Japanese government and the National Health Insurance (NHI) system for patients with sepsis who require vasoactive support with endotoxemia or in whom a Gram-negative infection is suspected [70]. Since then, PMX hemoperfusion therapy has been used in Japan in more than 100,000 cases [71]. The results of several small trials were beneficial with respect to the impact noted on survival, hemodynamics, and the pulmonary function. Regrettably, this has not resulted in the development of high-quality evidence for the efficacy of endotoxin removal via PMX hemoperfusion. A systematic review was published in 2007 of the studies coming from Japan. It identified 26 publications from Japan that were relevant and that pointed to the favorable effects on MAP, dopamine use, PaO2/FiO2 ratio and mortality, even though there was poor overall quality noted about the studies [72]. 

### 5.1. Randomized Controlled Trials of Endotoxin Adsorption Therapies

The results of randomized controlled studies (RCT) with endotoxin adsorbers performed in Europe and North America were inconclusive. Improved hemodynamic status and cardiac function was found in a group of 17 patients allocated to a single polymyxin B hemoperfusion (PMX HP) treatment, compared to the control group [73]. No differences in organ function or mortality were observed.

In the EUPHAS trial conducted in 10 Italian ICUs, two sessions of PMX HP performed in 34 patients resulted in significantly improved hemodynamics and organ dysfunction and reduced 28-day mortality [74]. The study suffered from premature termination by the ethics committee, when a statistically significant reduction in mortality was reached. This was criticized for the questionable statistical method used, the small number of enrolled patients, and the increased risk of type I errors [75,76]. 

An opposite result was found in the ABDOMIX trial conducted in 18 French ICUs. A non-significant increase in mortality and no improvement in organ failure was demonstrated in a group of 119 patients with peritonitis-induced septic shock after surgery who were subjected to two sessions of polymyxin B hemoperfusion [77]. A critique of the study included the lower 28-day mortality observed in the control group than reported in other studies with similar cohorts, a significantly higher rate of RRT in the PMX HP group than in the control group, and the fact that only 68% of the treated patients had completed two sessions of PMX HP [78].

The EUPHRATES trial was expected to give definitive evidence for the efficacy of endotoxin removal. The design of the trial used a theragnostic approach and included measuring endotoxin for patient enrollment. A simulated treatment in the control group was used to keep the treatment blind. The trial was the largest so far with 450 randomized septic shock patients, of which 224 were in the polymyxin B hemoperfusion group vs. 226 in the simulated treatment group. Only patients with EAA ≥ 0.6 were included. The result of the study was inconclusive, because the primary end-point of mortality at 28 days in the PMX-B HP group was not reduced [79]. A possible explanation for the negative result of the trial could be insufficient reduction in the endotoxin burden, which was supported by the observation that the PMX HP treatment did not result in significant reduction in EAA, especially in patients with a high baseline EAA [79]. This hypothesis was supported by the study on the relationship between EAA and endotoxin concentrations, which found that when EAA was higher than 0.9, the lipopolysaccharide concentration increased exponentially to the high non-measurable values [80]. It was speculated that when the level of EAA is above 0.9, the amount of endotoxin in the circulating blood may be very high and it may exceed the binding capacity of the adsorber. In addition, it is likely that a significant sequestration of endotoxin can occur in the extravascular compartment, which greatly heightens the total amount of endotoxin in the body [59]. Based on these premises, a post-hoc analysis of the EUPHRATES trial was performed and only patients with EAA ≥ 0.6–0.89 were included. A third of the 194 patients had an intra-abdominal infection, and another third had a lung infection. Gram-negative infections were present in only 18% of patients, Gram-positive in 27%, and cultures from 31% of patients showed no growth of bacteria. The 28-day mortality, adjusted for the APACHE II score and baseline MAP, was significantly lower in the PMX HP group, although in the unadjusted analysis the difference was not statistically significant. It was suggested that when strict patient selection criteria are applied, the use of PMX is associated with favorable changes in mean arterial pressure, ventilator-free days, and mortality; these findings should be considered as hypothesis-generating [59]. A study called TIGRIS (ClinicalTrial.gov Identifier NCT 03901807) is currently underway to validate this result.

### 5.2. Systematic Reviews and Meta-Analyses

Systematic reviews of publications on polymyxin B hemoperfusion in septic/septic shock patients yielded disparate results, depending on the inclusion strategy of eligible trials [72,81,82,83] and the statistical method used [3,84]. Earlier studies, mostly small or non-randomized, and representing a low quality of evidence, were in favor of using PMX HP treatment, whereas the results of RCTs published after 2010 did not show a beneficial effect [3]. The most recent meta-analysis that included 13 studies suggested that patients with sepsis and septic shock benefit from PMX HP in overall mortality compared with conventional medical therapy [84]. The authors speculated that a reduction in endotoxin levels and an improvement in the hemodynamics by the PMX HP treatment could contribute to improved survival. As opposed to previous publications, it also found that when subgroup analysis was based on disease severity stratified by APACHE II scores, then the mortality of patients with less severe sepsis, i.e., with an APACHE II score lower than 25, appeared to be significantly lower after PMX HP treatment.

An analysis of the data from EUPHAS 2, an international, multicenter registry, showed a favorable outcome in patients treated with PMX HP, similar to the EUPHAS study [85]. The data from the Japanese patient database showed better outcome in a cohort of patients with septic shock and kidney dysfunction that necessitated renal replacement treatment, suggesting that there was a benefit from the PMX HP therapy in a more severe population of patients [86]. On the contrary, patients who had open abdominal surgery for perforation of the lower gastrointestinal tract and who required vasopressor support were not found to have any survival benefit with a relatively low mortality of 16.3% in the control group [87].

## 6. COVID-19 and Endotoxemia

The presence of endotoxemia was demonstrated in patients with COVID-19 pneumonia [88,89]. Lipopolysaccharide was speculated to be of lung [89] or intestinal origin [88] and may have contributed to the pathogenesis of COVID-19. Of note, the presence of bacterial DNA in serum was detected in 49 of 50 samples from 19 COVID-19 pneumonia patients using the 16S rDNA sequencing method [88]. Endotoxemia was also observed in 75% of 92 critically ill patients with COVID-19, with only two patients having positive blood cultures for Gram-negative organisms [90]. Significant endotoxemia was found in a cohort of 22 patients with COVID-19-related pneumonia, together with increased serum zonula occludens-1 levels, a marker of the integrity of the intestinal paracellular barrier, implying that there was intestinal barrier dysfunction and supporting the speculation that bacterial translocation from the gastrointestinal tract may complicate severe COVID-19 and may contribute to the cytokine storm [88,91]. In another study, the co-existence of low-grade endotoxemia with enhanced levels of zonulin in patients with COVID-19 was observed, as well as an association with thrombotic events [92].

Only case reports and case series describing the effects of endotoxin removal in patients with COVID-19 have been published to date; therefore, their results should be treated with caution. In a case series of 12 patients from the EUPHAS 2 registry with COVID-19 and endotoxic shock due to a secondary infection, the PMX HP treatment was associated with organ function recovery, hemodynamic improvement, and a reduction in the EAA level [93]. Clinical improvement was observed after endotoxin adsorbent therapy in six critically ill patients with COVID-19 and an elevated EAA level. All six patients survived [94]. PMX HP performed in 12 patients with COVID-19 resulted in a decrease in disease severity in 58.3% of the patients on day 14 after the first treatment. It is noteworthy that a high frequency of clotting of the adsorber was observed [95].

## 7. Investigating Aspects of Endotoxin Removal

### 7.1. Timing of the Initiation of Endotoxin Adsorption

Non-randomized studies that compared an early vs. late initiation of the PMX HP treatment in patients with septic shock found better survival or reduced catecholamine requirements in the early treatment group. The initiation of PMX hemoperfusion within 6, 8, or 9 h after the administration of catecholamine or the diagnosis of septic shock resulted in a more favorable outcome compared to a later initiation [96,97,98]. According to the results of these studies, PMX HP therapy should be performed as early as possible in patients with septic shock, and a delay in PMX HP therapy may contribute to increased mortality [97].

### 7.2. Extended Endotoxin Adsorption Treatment

The recommended period for PMX HP treatment is 2 h. In studies where the time of treatment ranged from 8 to 24 h, there were improved hemodynamics and improved pulmonary oxygenation, but no improved mortality was observed [99,100,101].

### 7.3. Endotoxin Removal Treatment Guided by Measuring the Endotoxin Level 

In 11 patients diagnosed with postsurgical sepsis and who had a high EAA (≥0.6), when the PMX HP treatment was performed and repeated every 24 h until the EAA was low (<0.4), all patients survived until the 28-day follow-up [102]. These findings are similar to an observation from the post-hoc analysis of the EUPHRATES trial. A trend toward lower mortality and a significant increase in ventilation-free days was found in patients with septic shock and a greater than median reduction in EAA on day 3 after the PMX HP treatment. The same was true for patients who achieved an EAA of less than 0.65 on day 3 [103]. The authors of the study suggested that the dosing regimen of PMX therapy should be tailored according to measured endotoxin levels and/or patient’s clinical response, but this hypothesis needs to be validated in a prospective study [103].

## 8. Conclusions

Adjuvant therapies have, at most, only a supportive role in the treatment of septic shock. Nevertheless, antibiotics, controlling the source, and organ support remain the mainstays of treating sepsis and septic shock. Endotoxin removal therapy, guided by the blood endotoxin level and applied early in the course of septic shock has the potential to improve organ function and improve survival. The adverse effects of endotoxin in the circulation support the timely removal of endotoxin in the course of sepsis, before the vicious spiral of progression to septic shock, multi-organ failure, or death occurs. Unfortunately, this has not been proven in large, randomized studies. Blood purification therapies need further clinical evaluation and their place in the therapy of sepsis/septic shock has not yet been determined.

## Figures and Tables

**Figure 1 jcm-11-00619-f001:**
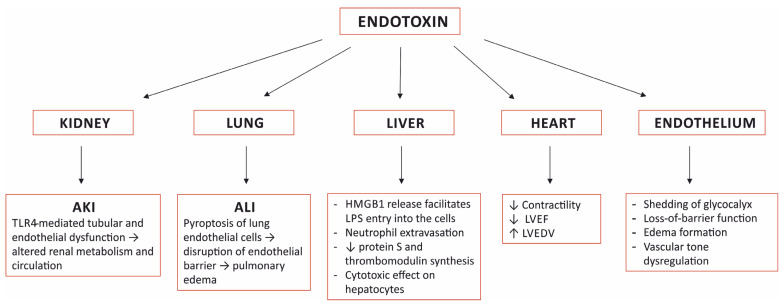
Selected organ damage induced by sensing endotoxin. AKI—acute kidney injury; ALI—acute lung injury; TLR4—toll-like receptor 4; HMGB1—high mobility group box-1; LPS—lipopolysaccharide; LVEF—left ventricular ejection fraction; LVEDV—left ventricular end diastolic volume.

## Data Availability

No new data were created or analysed in this study. Data sharing is not applicable to this article.

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
