# Peer review of "The Rationale and Current Status of Endotoxin Adsorption in the Treatment of Septic Shock"

_jcm, 2022, doi:10.3390/jcm11030619_

Round 1

Reviewer 1 Report

Dear Author,

Congratulations on your comprehensive work! Athough the paper itself is not a systematic review, it is a comprehensive, well-organized article that elucidates the topic of endotoxin absorption, starting from the very beginning, giving the accurate base to a reader that may not be familiar with the theme. The significance of the subject is huge, since it might possibly play an important role in intensive therapy in septic patients with infections caused both by Gramm-negative and Gram-positive bacteria. It was truly a pleasure  to read your work!

Author Response

I would like to thank the reviewer for taking the time and effort necessary to review the manuscript. I sincerely appreciate valuable comment and classification of the manuscript as very good.

Reviewer 2 Report

This is a comprehensive narrative review devoted to the problem of blood purification since the appearance of the method and up to time beings.The publication may make sense particularly for medical students, residents or other neophytes in a field of blood purification therapies. Unfortunately, last decade elsewhere some similar reviews and with the same conclusions have been already published: "Blood purification therapies need further clinical evaluation and their place in the therapy of sepsis/septic shock has not yet been determined". To my best knowledge this is it the reason that downgrades the scientific and practical significance of the article reviewed. 

Author Response

I would lie to thank the reviewer for taking the time to read and review the manuscript. Indeed, there are other reviews dealing with the subject of endotoxin removal, but this fact underscores the importance of endotoxin in the pathophysiology of sepsis. This could be the reason why the Guest Editor of the special issue of JCM invited me to write this review.

Reviewer 3 Report

This is a very well constructed and written review. It takes the reader through background ideas and theories in logical fashion to the use of PMX for the removal of LPS. The assessment and description of the efficacy of PMX I believe is a fair reflection of current opinion. 

Major point

PMX has not been proven to be the easy solution for the management of sepsis.  The reasons for its failure to fulfil its initial promise are only lightly touched on.  Could there be dose related issues or perhaps removal of other substances (e.g. lipoproteins) affecting it's efficiency. 

One minor comment. PMX is not qualified in the text.

Author Response

I sincerely appreciate these well-thought comments. A detailed description of PMX and other endotoxin adsorbers was beyond the scope of the review and it was intentionally not covered in the manuscript. The subject of possible interference of LPS adsorption therapies with e.g. lipoproteins is very interesting and a pioneering paper on the use of LDL apheresis for plasma LPS purification has recently been published by French researchers. However little is known about significance of this method of treatment for clinical outcome and therefore it was not mentioned in the manuscript.

I thank the reviewer for highlighting the importance of dose related issues. I have added a concluding remark at the end of section 7.3 „Endotoxin removal treatment guided by measuring the endotoxin level” (line 412-414) accordingly.

Line 412-414: The authors of the study suggested that the dosing regimen of PMX therapy should be tailored according to measured endotoxin levels and/or patient’s clinical response, but this hypothesis needs to be validated in a prospective study.